# Development of a Basic Educational Kit for Robotic System with Deep Neural Networks

**DOI:** 10.3390/s21113804

**Published:** 2021-05-31

**Authors:** Momomi Kanamura, Kanata Suzuki, Yuki Suga, Tetsuya Ogata

**Affiliations:** 1Department of Intermedia Art and Science, School of Fundamental Science and Engineering, Waseda University, Tokyo 169-8050, Japan; k24@suou.waseda.jp (M.K.); suzuki.kanata@fujitsu.com (K.S.); ysuga@suou.waseda.jp (Y.S.); 2Artificial Intelligence Laboratories, Fujitsu Laboratories Ltd., Kanagawa 211-8588, Japan; 3National Institute of Advanced Industrial Science and Technology, Tokyo 100-8921, Japan

**Keywords:** deep neural networks, educational kit, robot middleware

## Abstract

In many robotics studies, deep neural networks (DNNs) are being actively studied due to their good performance. However, existing robotic techniques and DNNs have not been systematically integrated, and packages for beginners are yet to be developed. In this study, we proposed a basic educational kit for robotic system development with DNNs. Our goal was to educate beginners in both robotics and machine learning, especially the use of DNNs. Initially, we required the kit to (1) be easy to understand, (2) employ experience-based learning, and (3) be applicable in many areas. To clarify the learning objectives and important parts of the basic educational kit, we analyzed the research and development (R&D) of DNNs and divided the process into three steps of data collection (DC), machine learning (ML), and task execution (TE). These steps were configured under a hierarchical system flow with the ability to be executed individually at the development stage. To evaluate the practicality of the proposed system flow, we implemented it for a physical robotic grasping system using robotics middleware. We also demonstrated that the proposed system can be effectively applied to other hardware, sensor inputs, and robot tasks.

## 1. Introduction

### 1.1. Background

Research and development (R&D) for combining robotics and machine learning is currently being actively pursued. With increasing applications of robots outside of traditional manufacturing, they are now required to perform various tasks in practical situations autonomously. To develop systems that are robust to changes in dynamic environments, R&D of learning-based robot manipulation systems has garnered much attention. In particular, research on the use of deep neural networks (DNNs) has produced excellent results in many fields [1,2,3,4]. As DNNs can handle raw high-dimensional data without complex preprocessing, they can be employed for various processes in robotic systems, such as path planning [1,4,5], object pose estimation [2,6], and robot motion prediction [3,7,8,9,10].

However, robotics is rooted in the field of mechanical engineering, while DNNs are rooted in information technology engineering. Therefore, it is difficult to develop human resources that are familiar with both fields. To conduct R&D for robotic systems using DNNs, basic knowledge and skills of both fields are required; however, resources for beginners to acquire such basic knowledge and skills currently do not exist. Furthermore, existing robotics technologies and DNNs have not yet been systematically integrated. Additionally, open common frameworks for easy reuse, exchange, and extension of knowledge have not been developed. To increase the user base of these technologies, existing products and services must be improved.

### 1.2. Purpose and Target of this Study

A dedicated introductory kit is required to educate people who can use both DNN and robotics technologies. In many cases, engineers acquire skills through trial and error using tutorials of the kit as a guide. In this study, we developed a basic educational kit for a robotic system with DNNs. We publish all the information on our system enabling users from all backgrounds to learn through practice. Our goal is to educate users on both robotics and DNNs. We aim to enable users to perform a series of development procedures for DNN robotics applications (training data collection, model training, and task execution on real robots) by themselves.

The main target users for this kit are beginners, such as students assigned to university laboratories or new employees assigned to a DNN robotics application development team. In particular, we target users who have some understanding of either DNNs or robotics and want to learn the other. Our educational kit is assumed to be used for self-studying.

Our developed educational kit comprises a robotic system and a manual explaining how to use the system so that beginners can experience system development by operating a physical robot. To clarify the learning objectives and important parts of the basic educational kit, we conducted a use case analysis and divided the work process into the three steps of data collection (DC), machine learning (ML), and task execution (TE). To evaluate the practicality of the proposed system flow, we applied a hierarchical system flow of these three steps to a robot grasping application using a physical robot and robotics technology (RT)-middleware [11]. The goal of this application was to have a robot arm grasp an object no matter where the object is placed; therefore, the paper does not elaborate on the details of the accuracy of the grasping task. Our system uses RT-components (RTCs) to control the robot arm and a camera, allowing it to be independent of specific hardware. This allows users to easily develop other robot applications that reuse the same components.

### 1.3. Contribution

As mentioned above, our motivation is to provide beginners on a development team with self-studying tools that are ready to use in their own environment. Our contributions are as follows:We proposed a system flow of a basic educational kit for DNN robotics application.We developed a DNN robotics application with RT-middleware and the proposed system and demonstrated some of its educational scenarios.We published the developed kit along with a tutorial for beginners.We showed some examples of extended robotics applications based on our educational kit.
The proposed educational kit’s target is system integration of the DNN and robotics fields, and to the best of our knowledge, there are no educational kits for DNN robotics applications with tutorials. Our study is the first step to lowering the entry barrier to the fields of DNNs and robotics, and it is expected to contribute to the development of these fields.

### 1.4. Paper Organization

The remainder of this paper is organized as follows. In Section 2, with reference to some related works, we describe the requirements of the basic educational kit for robotic systems with DNNs. Then, we describe the proposed system architecture in Section 3. In Section 4, we describe a robot grasping system developed using RT-middleware and a DNN based on the proposed flow. Then, in Section 5, we present the implementation of the application with a physical robot arm. In Section 6, we discuss the expansion of the developed system to other hardware, sensor inputs, and robot tasks. In Section 7, we discuss the effectiveness, evaluation, and limitation of the developed educational kit. Finally, Section 8 concludes the paper.

## 2. Requirements of Basic Educational Kit

In this section, we introduce some related studies on educational kits and investigate development support tools for both DNN and robotics fields. We also define the requirements for our educational kit in terms of teaching materials.

### 2.1. Educational Robots

Several educational robots have been proposed, not limited to DNN robotics applications. S. Evripidou et al. have investigated and analyzed existing platforms of educational robotics and their learning outcomes [12]. R. Burbaite et al. proposed a framework that allows teachers to educate students on computer science algorithms and programming using multiple robots [13]. I. Plauska et al. proposed the use of educational robots to encourage learning about IoT in college classes [14].

Previous studies have focused on robot development or other teaching targets, and no teaching materials have been integrated with DNN and robotics applications. In addition, no manuals or tutorials that support all procedures have been published. We developed an educational kit with a component-based system design using RT-middleware. This system design allows the user to develop advanced applications by expanding the basic educational kit. Details of the system implementation are given in Section 3 and Section 4, and examples of advanced applications are presented in Section 6.

### 2.2. Development Support Tools for Robotics and DNNs

There are some easy-to-use online services and tutorials on robotics and machine learning. In mechanical engineering, “robotics middleware" has emerged to efficiently construct robotic systems. Examples of robotics middleware include the robot operating system (ROS) [15], the open platform for robotic services (OPRoS) [16], and RT-middleware [11]. These software platforms are designed for the component-based development of robotic systems, where each developed component performs simple data processing, and the overall system is constructed by combining multiple components. Although the development of robotic systems requires a wide range of expertise and skills, the RT-middleware allows users to reuse developed components or share development flows with multiple developers. In addition, by using RT-middleware, a robotic system can be developed at a relatively low cost without extensive technical knowledge and skills. Thus, the robotics middleware is suitable for the basic educational kit for the robotic system.

In the machine learning field, there are certain libraries that make it easy to build neural networks, such as Tensorflow [17], Pytorch [18], Chainer [19], and Keras [20]. These libraries support most types of DNN layers, and developers do not need to implement backpropagation calculations for the training phase of a DNN. Using these libraries will simplify the introduction of machine learning processes into the kit. Additionally, the software can be used through web-based programming tools that do not require installation [21,22]. These tools are easy to use because they do not require an initial development environment to be built.

Despite the abundant support software available for each, existing robotics and DNN techniques have not yet been systematically integrated. One reason for this is that DNN-based robotic systems require training data and models to be developed to suit the particular task of the developer. This is especially the case in physical applications that are difficult to simulate. Dedicated systems are difficult to develop, and few studies have been published as open-source [23,24]. No package has been presented that is reproducible and reusable for the entire system developed, including DNNs and robotic systems. This poses an inconvenience for developers and difficult for beginners. Therefore, we considered these issues in the development of our educational kit, and we have developed the first educational tool for DNN-based robotic systems, using DNN and Robotics support tools, respectively.

### 2.3. Requirements

To design an effective educational tool, it is necessary to define the design guidelines of the tool. In general educational tools [25], the educational content and structure are of prime importance and are required to be easy to understand, convenient, practical, inexpensive, but beneficial. However, we do not consider the price because our educational kit to develop in this study is not intended to be a commercial product. Based on the above guidelines, we define the following three requirements for our educational kit.

Requirement (1):Easy to understandRequirement (2):Provides experience-based learningRequirement (3):Applicable to many areas

We design our educational kit for DNN-based robotic systems with the above requirements in mind. Since these requirements are for a general education kit, we need to adjust the above for the educational kit for DNN-based robotic systems. We analyze the requirements from the viewpoint of system integration of DNNs and robotics.

First, we set the requirement (1) because our kit should support an introduction to the R&D of DNN-based robotic systems. To design an educational kit suitable for learning basic knowledge and techniques, we focused on ease of understanding through simple system design and detailed descriptions. Our developed kit supports users to learn the basics of robotics and DNNs through the experience of system development using RT-middleware.

Next, we set requirement (2) because we believe that learning practical knowledge and techniques is as important as theoretical learning from, e.g., books. Thus, the educational kit must have an easy-to-use implementation for beginners. By using RT-middleware, our developed kit supports users to acquire the basics of robotics and DNNs through the experience of system development (experience-based learning).

At last, we set requirement (3) because our kit should be usable as a baseline for system design when a user develops their own system. By constructing a system that can be easily applied to a wide range of applications, it is useful for actual R&D of DNN-based robotic systems.

## 3. Proposed System Flow

In this section, we describe the hierarchical system flow for the R&D process with DNNs. To meet the requirement (1), we developed the system based on the simplest possible system design. By analyzing the actual R&D process, we determined the packages of each layer so that they can be understood intuitively (Section 3.1). After then, by combining packages in a hierarchical manner, it is possible to create a sequential data flow that is easy to understand and implement for beginners (Section 3.2).

### 3.1. Analysis of R&D Process of DNN-Based System

Learning objectives for beginners must be clarified to develop a basic educational kit. We focus only on the important parts for development, not emphasizing the technical or detailed parts that interfere with the learning process for beginners. This is based on the concept of experience-based learning, where it is important for users to learn the general flow of development through actual experience with the help of a basic educational kit.

In this subsection, we report a use case analysis to determine which R&D processes should be included in the educational kit. The activities of a DNN-based system can be divided into the preparation of DNN models and execution of the task using DNNs. In the preparation phase, data are collected and preprocessed for training the model. The training parameters and model of the DNN are tailored to the target task. In the task execution phase, the prediction process is executed using the input data, trained weight parameters, and model.

Preparation of DNN model: The functions required for DNN model preparation can be summarized as follows: (a) data collection, (b) data preprocessing, (c) learning parameter determination, (d) DNN structure determination, and (e) training function of the DNN. Figure 1 shows the requirements diagram of the DNN model preparation. In data collection (a) and data preprocessing (b), the data necessary for the training of the task are retrieved/stored/created. DNN training usually requires a large number of datasets. In fields such as image recognition, publicly available datasets, such as ImageNet [26] and MNIST [27], can be used. However, existing datasets do not always provide the requirements for R&D, which means that the user must prepare their own dataset. Particularly in the field of robotics, developers almost always have to generate their own datasets. Many efforts have been made in the collection and preprocessing of training data in R&D, and these elements are very important in the development flow.

In the training function of the DNN (e), we retrieve the results using the training data, training parameters, and DNN model. The training parameters and model of the DNN are regarded as the tunable parameters (c,d) of the system. Based on the results, if the learning is judged to be successful, the weights of the trained DNN and its structure are passed to the next phase. In the case of learning failure, data must be recollected or the training parameters and structure of the DNN model must be re-selected.

Execution of a task using the DNN: The functions required for the execution of a task using the DNN are as follows: (f) construction of the DNN model from received it, (g) loading the trained weights, and (h) executing a task with the DNN. Figure 2 shows the requirement diagram for task execution using the DNN. In the next phase, tasks are executed using the DNN model and trained weights. The system constructs a DNN from the loaded model and trained weights. By inputting the data for prediction, the model outputs the information required to execute the task.

The process described above constitutes the minimum requirements for R&D with DNNs. Through experience, users can learn practical knowledge and skills. In addition, R&D with DNNs can be divided into separate steps, as described above. We packaged them individually so that the system can be extended to adapt to the user’s skill level. The user needs only to rebuild certain required or specialized parts for each step, thus avoiding the complexity of an educational kit as a development tool.

### 3.2. Overall System Flow

As each process of R&D for a DNN is at the general system level, we must configure the system architecture at the human activity level by combining the packages. Therefore, we constructed the system flow with the three steps: DC, ML, and TE. The correspondence between the functions (a)–(h) and these steps is illustrated in Figure 3. The execution of a task using a DNN and the preparation of the DNN model described in the previous section are connected by the model configuration and the weight parameter passing part. Furthermore, the preparation of the DNN model is divided into the DC and ML steps.

In the DC step, training data are collected and preprocessed so that the data can be efficiently processed in the next step. The training of the DNN requires large amounts of data to avoid overfitting. In the ML step, the training of the DNN is performed using data collected in the DC step to optimize the model’s parameter to the target task. The structure of the model and hyperparameters of the training process are determined for the target task. In the TE step, the target task is performed using the DNN model and trained weight parameters. The individual packages shown above were designed based on general DNN R&D, which allows them to be extended to a wide range of systems, as discussed in Section 6.3.

Figure 4 shows the flow of the proposed system, which has a hierarchical flow. The term ”hierarchical" in this context indicates that data flow from top to bottom. As no data travel back from lower to higher layer packages, the upper layer packages are not affected by development errors in lower layers. This system concept is also used in the detection of system faults [28].

A system that is divided and constructed following a development flow is suitable for educational kits as it is easily understandable by users. Additionally, because the system is run sequentially, it is easy to identify the cause of problems and modify the package when they occur. The proposed system flow can be applied to the development of a DNN-based robotic system, which was the purpose of this study. In the next section, we demonstrate the effectiveness of our educational kit through the implementation of a practical application using the proposed system flow.

## 4. Application to Robotic Grasping

As a basic educational kit for robotic systems with DNNs, we developed a vision-based robotic grasping system using the hierarchical system architecture described in the previous section. To meet requirements (2) and (3), we chose software that can be conveniently used by anyone. By developing a reusable and extendible system, users can easily customize the system for their environment. We used RT-middleware [29] for the robotics software and Tensorflow [17] and Keras [20] for the DNN software. These are rich in tutorials that make our developed kit more convenient. Moreover, our system configuration allows for easy expansion of the system, such as replacing hardware, adding sensors, and applying the system to other robot tasks.

We chose the robotic grasping task because object manipulation by a robot arm is an important application in robotics. The robotic grasping task is easy to introduce DNNs to, which has shown its good performance in computer vision. In this task, the DNN predicts the grasping position from input images. Here, we can consider the motion plan and image recognition parts separately. The two main functions of this task are as follows:Estimation function: the DNN estimates the graspable position for the target object from an image received by a camera.Grasping function: the robot arm moves its gripper to the estimated graspable position and grasps the object.
These functions are realized by combining robotics and DNN technology, so the task is suitable for our educational kit.

Through the picking task, DNN is expected to automatically extract the image features necessary to perform the task from the input image and output the grasping position end-to-end. The image features required for task execution are the geometric features on the image that are associated with the robot’s gripper and the object. Since the DNN is difficult for humans to understand what is processing inside, the engineers of DNN-based robotics systems need to learn how to construct the model through trial and error of DC, ML, and TE steps.

We implemented the DC, ML, and TE steps with the software to be easy to use even for beginners. In the DC and TE steps, RT-middleware was used. Using RT-middleware, users can develop robotic systems without specialized knowledge of the required technologies or hardware. In the ML step, we prepared only a sample script of a DNN, which is constructed using machine learning libraries. The hardware selected for implementation was the minimum required for system operation, including a robot arm as the manipulator, a camera as the sensor, and an object to be grasped.

### 4.1. RT-Middleware

In the parts of the system development using robotics, RT-middleware was used for implementation. RT-middleware with support for OpenRTM-aist, which is a tool that enables component-based development, was used to control the robot and develop the system [29].

The functional elements required to construct a robotic system are divided into units called RTCs. RTCs can be combined to build a robotic system. RT-middleware is designed to let the user connect components in the GUI. Because the user must be aware of the input/output data type of each component, this facilitates their understanding of the entire robot system. Thus, it is suitable for beginner use. Note that the proposed system can also be implemented with other RT-middleware, such ROS. In this study, we adopted RT-middleware from the viewpoint of ease of use and our development environment.

To maximize the reusability of the modules of the RT-middleware, we chose to use a common interface specification in the system. Even if the type of manipulator used is different, the upper-level component that sends instructions can be controlled by the same command. In our system, the RTCs for controlling the robot arm and camera are prepared, and common interfaces are defined for the control function of the robot arm and the camera. These interfaces allow us to develop the robotic system without depending on specific hardware. Therefore, users can implement our system using their own available hardware.

### 4.2. Data Collection Step

The purpose of the DC step is to collect data for DNN training. To train the DNN to predict a graspable position for the target object from an image, image and position attitude (XY coordinate and rotation angle) pairs are required. However, collecting the training dataset for robot tasks is resource-intensive in terms of time and funds [30,31,32]. For our educational kit, we developed the DC package so that it could collect the dataset automatically. The DC procedure is as follows:(1)The developer has the robot arm grasp the target.(2)The system repeats the following steps:(a)The robot moves to the initial position.(b)The robot moves the gripper to a randomly generated position attitude with XY coordinates and rotation angle (x, y, θ) within the workspace.(c)The robot places the object at the position attitude on a table.(d)The robot moves the robot arm to a specific attitude and then the camera captures an RGB image.(e)The system saves an RGB image and position attitude (x, y, θ) pair.(f)The robot moves to the generated position attitude again and grasps the object.
Figure 5 shows DC system operation flowchart. The only small cost to humans is the initial setup, after which the system automatically collects data until it has collected a specified amount. The collected images, their file names, and corresponding coordinates are saved as a set in a CSV file. In the ML step, by loading this file, it is possible to train on the data of camera images and position attitude pairs.

In the DC step, we implemented the package with RT-middleware to control the robot and other hardware. Figure 6 shows the RTCs in the DC step. The DC package consists of three RTCs: MikataArmRTC, WebCameraRTC, and ArmImageGenerator. ArmImageGenerator is an RTC developed for this DC package. It commands automatic data collection by operating the robot and saving RGB camera images and position attitudes (x, y, θ). The data acquisition flow is as described in the previous paragraph. The interface of this RTC consists of ManipulatorCommonInterface Middle and ManipulatorCommonInterface Common. These are common interfaces for the control functions of the robot arm. The former is a common interface for motion commands, and the latter is a common interface for commands used for status acquisition. In addition, it is necessary to deal with changes in the task that are caused by the hardware used for implementation, as the configuration of the hardware is highly compatible. Using the configuration function of RT-middleware, we can change the following parameters from the graphical user interface (GUI): initial attitude of the robot arm, attitude of the camera shot, grip open/close width, time between movements, waiting time during shooting, and range of the workspace for the robot arm.

### 4.3. Machine Learning Step

In the ML step, the system trains the DNN with the dataset collected in the DC step. The goal of the ML step is to obtain a model that can estimate a graspable position for the target object from an RGB image. We constructed the ML package with TensorFlow [17] and Keras [20], which are libraries for machine learning. Keras is a neural network API developed with a focus on enabling fast experimentation; it allows training models to be constructed by stacking layers. This makes Keras easy to learn and use; even users with no prior programming experience can readily and easily prototype.

Constructing a training model from scratch is a challenging task for first-time learners; therefore, we prepared a sample DNN model of a convolutional neural network (CNN) [33,34]. The prepared sample scripts were not intended to achieve grasping with high accuracy, but rather be modifiable by the user to improve the performance and modify the input/output to suit the intended purpose. In the published kit, the user can skip the model parameter tuning part by using the sample script. After running the sample script, the user will get the trained weights in HDF5 file format and the DNN structure in JSON file format. By providing the completed system as a teaching aid, the user can experience the development of a DNN-based robotic system in an easily interpretable manner. This service is important for first-time learners to understand the outline of the system development flow through practical hands-on experience.

### 4.4. Task Execution Step

Using the weights and DNN model obtained from the ML step, the TE step recognizes the graspable position for the target object from the input image. Then, it performs the grasping action. The system repeats the above process until the end flag is sent. The performance of our system is dependent on the model output accuracy in the ML step. Therefore, if the required task is not achieved in the TE step, it is necessary to change the structure of the DNN and/or hyperparameters in the ML step or to collect further data in the DC step.

As with the DC step, RT-middleware is used to control the robot and other hardware. Figure 7 shows the RTCs in the TE step. The TE package consists of three RTCs: WebCameraRTC, MikataArmRTC, and KerasArmImageMotionGenerator. KerasArmImageMotionGenerator was developed for the TE package to utilize Keras libraries, and this RTC uses the trained DNN to estimate the graspable position for the target object and commands the robot to grasp. The RTCs for controlling the robot arm and camera are reused from the DC step. Each RTC has a common interface that can be easily appropriated. As with the DC step, the connection between RTCs can be performed from the GUI. For implementation details, please refer to our publicly available software.

### 4.5. Manual and Tutorial

The developed kit comprises a DNN-based robotic grasping system and an operation manual for the system. This manual is published on a webpage, and all software used in the kit can be installed from the links provided in the manual. It is also possible to contact the provider of the educational kit through the comments section of the manual. For details of the manual, please refer to the Data Availability Statement.

The manual also includes a tutorial that describes what the developer should do at each step. The overall work described in the tutorial can be roughly divided as follows:Overview of the educational kitSoftware installationHardware settingsDetails of how to use the systemExecution of DC, ML, and TE steps

The user can experience the development procedure from scratch by sequentially performing the operations described in the tutorial. It is expected that the tutorial will lower the entry barrier for beginners and increase the number of developers.

### 4.6. Scenario

The educational scenario envisioned in this study is practical self-studying by the user. Its purpose is slightly different from the lesson-based education conducted at a university. The following are some examples of educational scenarios using our developed kit:Project-type self-studying for students assigned to university laboratoriesSelf-studying for new employees assigned to a company’s development team
If the users prepare a PC, webcam, and robot arm with a gripper, they can experience the development procedure of a DNN robotics application. Therefore, our developed kit is easy to introduce for self-training.

## 5. Implementation of Sample Robot Application

To verify the practical application of the system, we implemented a robotic grasping system using a physical robot and confirmed its task execution.

### 5.1. Hardware

The target task of this system was to reach and grasp a box placed on a table. Figure 8a shows the experimental setup. The size of the box selected as the target object for grasping was 95×45×18 mm. We implemented the system using Mikata Arm [35], which has six degrees of freedom (DoF), a two-finger gripper, and an RGB camera [36]. To control the robot arm and camera, we used the MikataRTC [37] and WebCameraRTC [38], respectively, which are both open-source. The RGB web camera was attached to the tripod above the arm, and Figure 8b shows an image captured by the camera. The hardware was kept in the same position during the DC and TE steps. This was to prevent the model accuracy from being reduced due to a change in the viewing angle of the camera.

### 5.2. Learning Experiment

We executed the DC, ML, and TE steps on the implemented system. In the DC step, we collected approximately 500 images in 2 h. All images were annotated with position attitudes, as described in Section 4.2. In the ML step, we trained the sample DNN model implemented by Keras. The DNN model consisted of four convolutional layers and three fully connected layers. The last layer had a linear activation, and the other layers had rectified linear unit (ReLU) activation. The input of the DNN model was the camera image resized to 64×64 pixels, and the output was a position attitude (x, y, θ) in the coordinate system of the robot. The loss function was the mean squared errors between the output of the DNN and target signals. We used 33% of the image dataset as the validation data. The model was trained using the Adam optimizer [39] over 3000 epochs. The batch size was 10. The training process took less than 3 h. The results of the learning curve are shown in Figure 9. Based on the loss and accuracy values, we judged the training to have converged. We used the weight parameters at the time of the 3000th epoch for the next TE step.

In the TE step, using the parameters obtained from the ML step, we constructed an image recognizer and verified whether the grasping task for the target object could be performed. The robot arm could perform grasping regardless of the position of the placed object (Figure 10). The success rate of the performed task was over 85%, as was the accuracy on the validation data. Note that the accuracy of DNN robotics applications largely depends on the number of datasets and execution environment of the application. Because our grasping task is simple, it was possible to train the DNN model with a small amount of data. This task is sufficient for beginners to experience development flow. From the above, it was confirmed that our developed application operates normally.

In this picking task, the middle layer of CNN learns geometric features of objects, such as edge, shape, the difference between the background and the object, and shading. To construct a well-performing model, the user learns how to collect training data that emphasize those image features through the implementation. As one example, it is known that the larger the variation of object positions and poses in the training images, the higher the generalization ability of the DNN model. For other detailed techniques, please refer to the survey on DNN [40] or tutorials on support tools [17,20]. They are the actual and useful techniques for the development of an actual DNN robotics application.

## 6. Expansion of Proposed System

In this section, we discuss the extendibility of the proposed system in terms of the requirement (3) described in Section 2. To meet requirement (3), the proposed system must be executable with low reimplementation costs, even in an individual user’s development environment. We considered three aspects of extension to other applications: (a) different hardware, (b) different sensor information, and (c) different robot tasks. Aspect (a) is necessary for the proposed system to be used by a wider range of users. Regarding aspect (b), it is preferable that the system can easily add new sensor information because of the characteristics of DNNs, which can handle a variety of sensor information in an integrated manner. For aspect (c), the system supports the development of other robotics applications that apply DNNs to processes other than recognition.

To verify the extensibility of the proposed system, we conducted three types of experiments. We applied the proposed system to another robot application by replacing the robot hardware (Section 6.1) and adding depth sensor information (Section 6.2). We also discuss the possibility of extending the system to support developing an end-to-end robot motion learning system (Section 6.3).

### 6.1. Hardware Replacement

First, we applied our developed system to the robotic grasping application with different robot hardware (aspect (a)). Our system uses RTCs to control the robot arm and camera, so it is independent of specific hardware. The RTCs have common interfaces, and we can easily develop another robot application by reusing these components. In this subsection, we show an example of replacing the robot hardware, which means replacing the RTC currently used in the system with another one. The replacement of RTCs is possible through the GUI of the RT-middleware. Users are only required to download the new RTC from a public repository and store it in a place where the system can access it. As described in Section 4.2, the parameters required for the robot grasping application can be adjusted from the GUI, so there is no need to edit the code or recompile the RTCs. Other detailed notes can be found in our manual.

To demonstrate that the hardware can be replaced in our system, we implemented the robotic grasping system using a different robot arm, OROCHI [41]. Figure 11a shows the experimental setup and Figure 11b shows the RGB image input to the DNN model. The robot arm has 6 DoF and an attached two-finger gripper with a hand-eye camera.

To control the robot arm, we replaced the MikataRTC with the OrochiRTC, which is a commercial RTC that supports OROCHI. Both robots have the same DoF and can be applied to the same system by only adjusting the parameters of the gripper open/close width, initial attitude, camera shooting attitude, and range of the workspace. The robot task was to grasp a box placed on the floor. The box was placed within reach of the robot arm. In this experiment, the RGB image captured from the hand-eye camera was used as the input of the DNN model.

The DC, ML, and TE steps were executed as described in Section 5.2. After the DNN training, the robot arm was able to perform grasping regardless of the position of the placed object (Figure 12). It was confirmed that operation of the system is possible even if the hardware is replaced, with only a slight change in system settings. Because the proposed system can be implemented in the user’s environment, the introduction and reimplementation costs of the system are low. This is important for beginners who are prone to mistakes in setting up a development environment.

Through the implementation of picking tasks using different hardware, users can learn how to replace RT components. When replacing components, it is important to organize the processing of each function and the input/output data types to be handled. For many robotic applications, adapting their specifications to the user’s operating environment is an important and difficult task, so our kit is expected to be useful in actual development.

### 6.2. Adding Another Sensor Information

Next, we added new sensor input to the robot grasping application (aspect (b)). Similar to hardware replacement, the RTCs allow the easy addition of sensor information to the proposed system. This subsection describes an example of adding a sensor input, i.e., a new RTC and its input/output, to the system. For building a new RTC, the templates of the RTC provided by RT-middleware can be used. If new inputs are added to the system, it is necessary to add ports to match the inputs in order to receive them. Using the functions provided by RT-middleware, this process can be easily achieved.

We set up another grasping task that could not be solved by the previous robot application and aimed to solve it by adding depth information to the system. The robot task at the TE step was set to grasp an object placed on a table that was covered with the same pattern as the surface of the target object. Note that, in the DC step, the training data were collected at the same table environment at the previous subsection. Figure 13a shows the experimental setup in the TE step. Since the pattern on the table interferes with the object recognition process, it was difficult to execute the grasping task using only the RGB camera image.

To address this issue, we adopted depth information as additional input from the sensor. As the RGB+depth (RGB-D) camera, we used RealSense r200 [42]. To control the RealSense r200, we replaced the WebCameraRTC with RealSense3D [43] and added depth information to the input. We extended the DNN to estimate the position and orientation of the object from the RGB camera image and depth image. Depth information was acquired as a matrix of the same size as the RGB image, and the input image to DNN was 64×64×4 pixels. Figure 13b,c show the examples of RGB image and depth image collected at the DC step. Other experimental settings were the same as described in Section 5.2.

After training the DNN with the acquired RGB-D images, the robotic arm became able to complete the grasping task by utilizing depth information (Figure 14). Through the implementation of a depth-informed picking task, users can learn how to configure DNN models and hardware according to the task execution environment. The addition of depth sensor information described here is a classic example of where the structure of a DNN must be tuned to the actual environment. The depth information is noisy compared to a normal RGB image, but it has more enhanced background difference information. By adding the input sensors with different characteristics, the model was optimized to extract the image features needed for the task from information with RGB images and depth images. With our developed kit, users can experience thinking about system requirements and DNN structure while employing a trial and error methodology in the actual environment and machine. Because users can construct applications without having acquired the specialized skills necessary for system design, users can conduct experience-based learning.

### 6.3. System Expansion to End-to-End Robot Motion Learning

Finally, we extended our system to an end-to-end robot motion learning system as an example of a more advanced robot application (aspect (c)). Because our system has independent development steps in a hierarchical system flow, it can be easily applied to different robot applications by changing the processing of each package. This subsection describes how to build an online motion generation application for a robot.

End-to-end robot motion learning is currently being studied as a promising approach to perform difficult robot tasks, such as manipulating flexible objects [3,44]. Although we used DNNs only for the recognition part, Refs. [3,44] used two DNNs, an autoencoder (AE) for feature extraction and a recurrent neural network (RNN) for motion learning. First, they trained the AE for extracting low-dimensional image features that represent the relationship between the object and robot arm in task manipulation from high-dimensional raw images. Then, they trained the RNN to estimate the next motion based on the previous image and robot state. By iterating these processes, the system can generate appropriate motions for the robot to perform its task online. To realize this system, we extended the educational kit by modifying the DC, ML, and TE steps.

Figure 15 shows the modifications of the hierarchical system flow. First, it is necessary to collect time series data consisting of sensory and motor signals during task execution as training data in the DC step. For this purpose, it is required to keep acquiring and storing the motion and image information while the robot is moving. We prepared RTCs to control the robot arm and camera to fit this task. Note that if the hardware was not changed from Section 5.1, the RTC would not require preparation. In addition, we separated the motion commands and information management using one RTC representing ArmImageGenerator in the grasping of Section 4.2. The flow of the ML step was almost unchanged, training the DNNs using the data collected in the previous step. However, the training data and types of DNNs used were different, so reconstruction of the training model was required. An AE and RNN can easily be built using libraries for machine learning, such as Keras or PyTorch. In the TE package, the robot generates motion while acquiring sensory signals (images) and motion information (joint angles) online. The RTCs controlling the arm and camera can be reused from the DC step, and the data acquisition process does not need to be redesigned. By adding a process that repeats robot control and data acquisition at an arbitrary cycle, online motion generation could be realized.

The application described in this subsection is based on the robot’s task experience collected in the DC step as training data, so it can be applied to various tasks. In fact, in another study by us [3,44], the above application was already implemented to learn a different robot task from that described above [45]. This was developed using our educational kit as a baseline for system design, demonstrating the effectiveness of the proposed system.

## 7. Discussion

In this section, we discuss the limitations and effectiveness of the proposed method in the research domain.

### 7.1. Effectiveness

As mentioned in Section 2, most related educational materials are limited to either DNNs or robotics, and none are published with tutorials. As our developed kit implements DC and TE steps using RT-middleware, it is easy to apply in the developer’s own environment. This point is important for materials related to system integration, and our study is the first step to lowering the entry barrier to the fields of DNNs and robotics. Because the materials proposed in previous studies have different targets from ours, we believe that they play a complementary role to ours. By using both types of materials, users will have a better understanding of both the DL and robotics fields.

We believe that our developed kit can work as an introduction tool for DNN robotics application development, such as MNIST, which is used as a DNN tutorial tool. The educational kit with a tutorial helps increase the number of users. As the number of users increases, R&D projects will be released as open-source, and the field will develop further. It is important to operate real hardware for the development of DNN robotics applications, which is different from DNN tutorial tools. Our developed kit supports the connection of these fields. Although the contribution of this paper is not a technological advance, providing a development experience will contribute to the development of the fields.

### 7.2. Evaluation

The proposed system allows users to easily experience the DC, ML, and TE steps in the R&D of a DNN-based robotic system. GUI tools are provided for both RT-middleware and Keras, so the user can develop robotic systems without any technical expertise. By creating RTCs that have a common interface, it is possible to implement the system independently of specific hardware. In addition, the hardware required to operate the system was minimized to only three components: a robot arm (manipulator), camera (sensor), and box (target object). The user can construct the system with their own hardware without purchasing specific items. Because our system is a highly practical implementation that even beginners can easily use, the developed system can be said to meet requirements (1) and (2) specified in Section 2.3.

In addition, Section 6 showed that our developed system has high versatility and extendibility owing to the ability to develop each step (DC, ML, and TE) individually. Moreover, using RT-middleware in our system allows easy hardware replacement or the addition of other sensors. From this, the developed system can be said to meet requirement (3) specified in Section 2.3. The three requirements should be evaluated quantitatively by a sufficient number of examinees. We believe that such an evaluation can be conducted by preparing development issues for each requirement and implementing a feedback function with an evaluation input function.

However, from the beginner teaching materials for handling system integration, it is difficult to make a strict comparison and evaluation of the three requirements. For quantitative evaluation, it is necessary to prepare comparison methods and use beginners as examinees to experience the full development procedure under the same conditions [46]. This would require a large time and human costs and is a unique issue in terms of the basic educational kit for development procedures with real hardware. One of the important contributions of this study is the development and publication of a basic educational kit for system integration of robotics and DNNs. To the best of our knowledge, there is no educational kit for DNN robotics applications that supports data collection, learning, and robot execution functions. We are waiting for feedback on the published kit and plan to improve the materials and compare them with other system designs.

### 7.3. Limitation

There are two main limitations to our developed kit. First, our developed kit can be used only with real hardware. Of course, the proposed system can be applied to simulations, but the manual does not describe how to use the kit in a simulator environment. This point prevents introduction to beginners who want to try the kit only on a PC.

Second, the manual does not support the expansion part of this educational kit in detail. As described in Section 6.3, the user can develop DNN robotics applications based on our kit, but because there are a wide variety of applications, it is difficult to follow all of them. The manual is intended only for development flow as a basis of DNN robotics application, and further enhancement of the materials is required for subsequent application development.

## 8. Conclusions

We developed a basic educational kit that comprises a robotic system and a manual to introduce people to robotic system development using DNNs. The key targets demographic for our kit were beginners who are considering developing a robotic system using DNNs. Our goal was to educate such people in both robotics and DNNs. To achieve this goal, we set the following three requirements: (1) easy to understand, (2) employs experience-based learning, and (3) applicable in many areas.

For the requirement (1), we developed a simple system design for users and prepared a manual as a detailed description. The educational kit should clarify learning objectives and make the development flow comprehensible for beginners. Thus, we analyzed the process of R&D with DNNs, and we proposed a hierarchical system flow that divides the overall system into three steps. For the requirements (2) and (3), we implemented the system with suitable software that can be conveniently used by everyone. We created individual functions for the three steps of R&D with DNNs, allowing beginners to experience the system development methodology easily.

To verify the practical application of the system, we developed a robot grasping system using a DNN based on the proposed system flow and the manual for our system. Our application in a physical robotic system was implemented using RT-middleware. This system constitution using the RT-middleware package allows easy extension to other hardware, sensor inputs, and robot tasks. In Section 6, we presented some examples of other robot applications. This paper provided concrete examples to supplement basic knowledge in DNN-based robotic development. Through the implementation of picking tasks, the users can learn how DNN works and how to improve it. In addition, the users can learn ways to apply robot applications to their development environments.

Our developed kit gives beginners the experience of developing a DNN-based robotic system while operating a physical robot as we believe that practical experience enhances comprehension of theoretical knowledge. The proposed system is just one approach to extending the knowledge and techniques of robotic systems using DNNs. In terms of encouraging increased use of the proposed system, we plan to develop additional functions that can customize the system for matching the user’s environment and share that information.

## Figures and Tables

**Figure 1 sensors-21-03804-f001:**
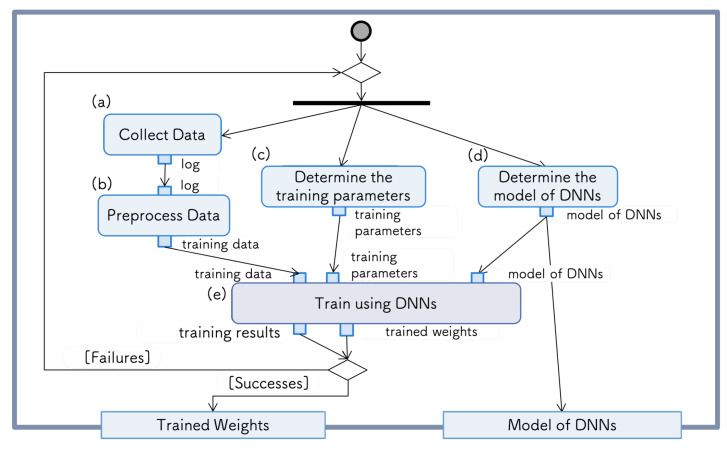
Process of DNN model preparation.

**Figure 2 sensors-21-03804-f002:**
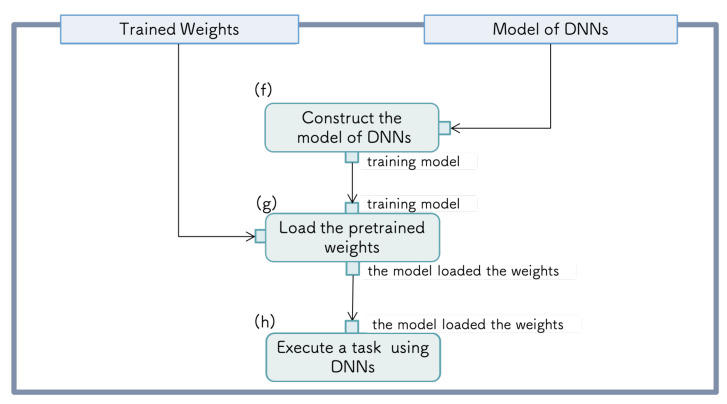
Process of the task execution using a DNN.

**Figure 3 sensors-21-03804-f003:**
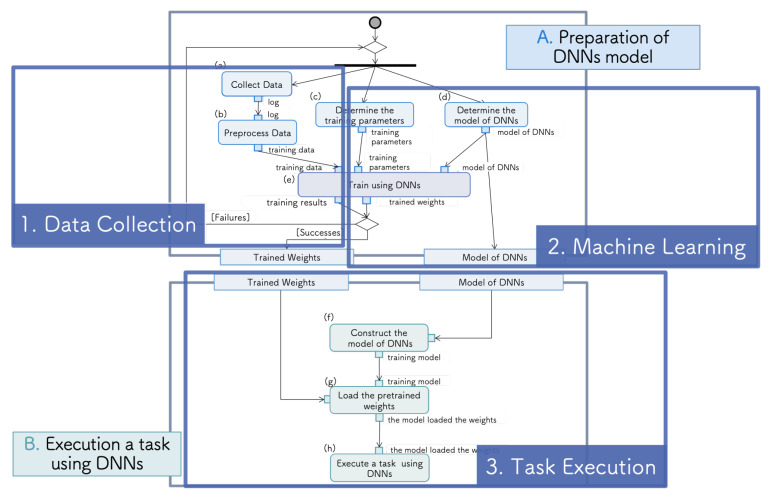
Correspondence between the typical method and proposed system model.

**Figure 4 sensors-21-03804-f004:**
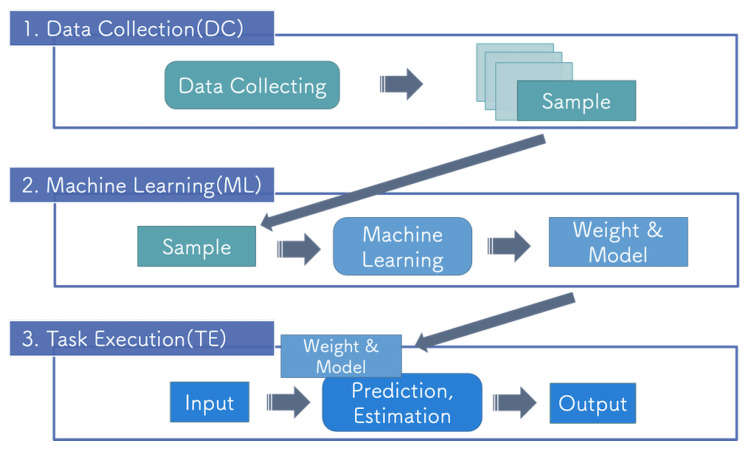
Hierarchical system architecture.

**Figure 5 sensors-21-03804-f005:**
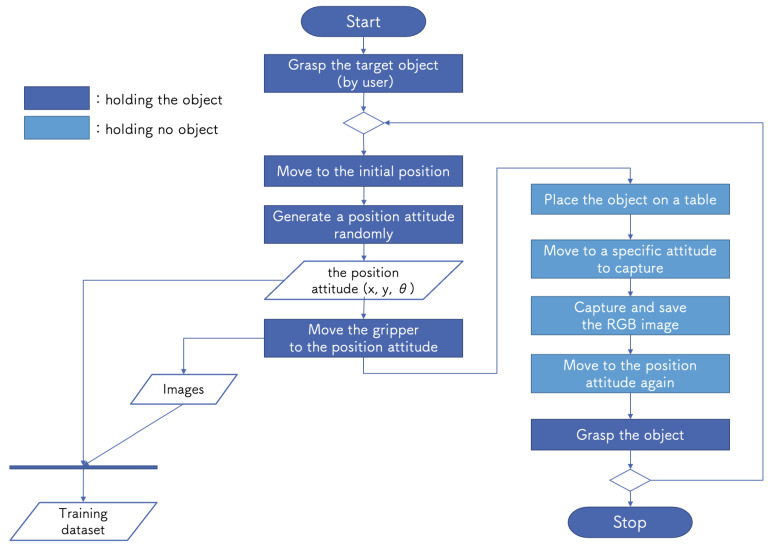
DC system operation flowchart.

**Figure 6 sensors-21-03804-f006:**
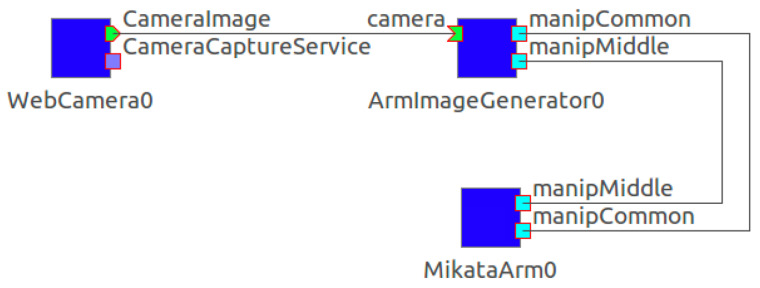
RTCs in the DC step.

**Figure 7 sensors-21-03804-f007:**
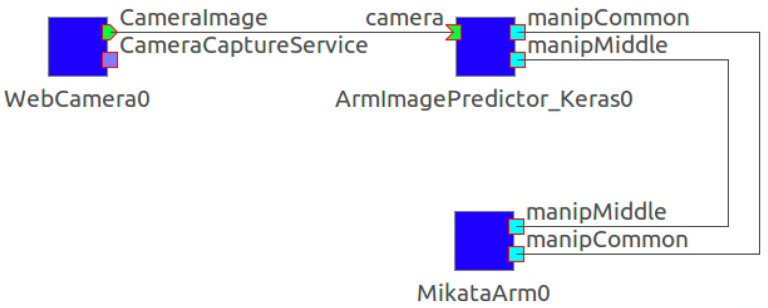
RTCs in the TE step.

**Figure 8 sensors-21-03804-f008:**
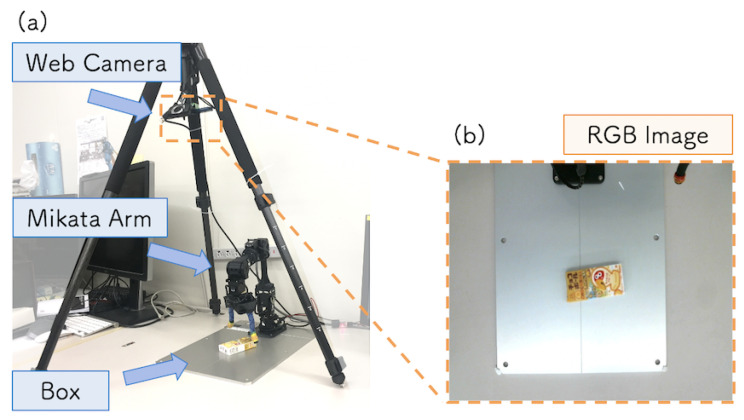
(**a**) Experimental environment setup. (**b**) RGB image captured by the camera.

**Figure 9 sensors-21-03804-f009:**
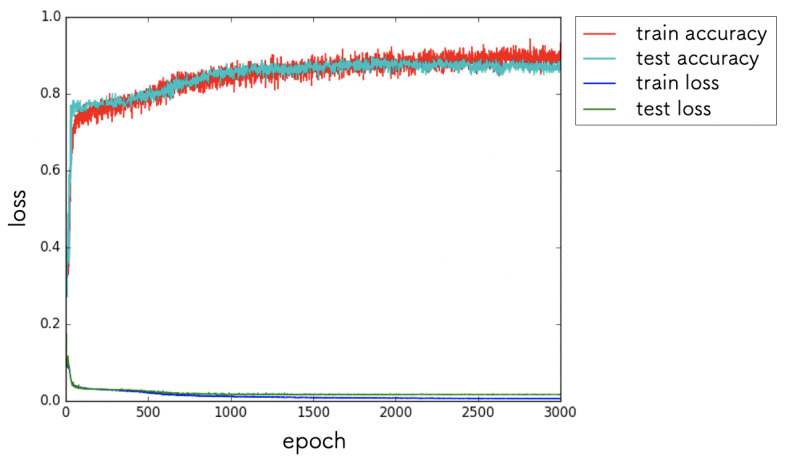
Results of the learning curve.

**Figure 10 sensors-21-03804-f010:**
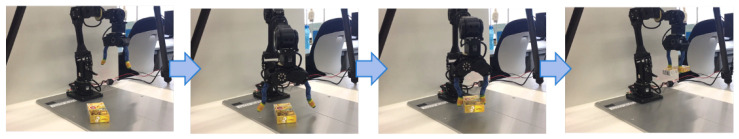
Example of successful grasping.

**Figure 11 sensors-21-03804-f011:**
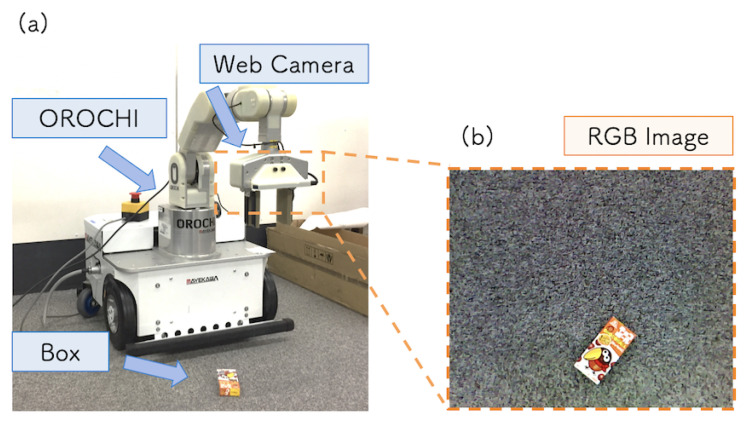
(**a**) Experimental environment for the robot grasping application with hardware replacement. (**b**) RGB image input to the DNN.

**Figure 12 sensors-21-03804-f012:**
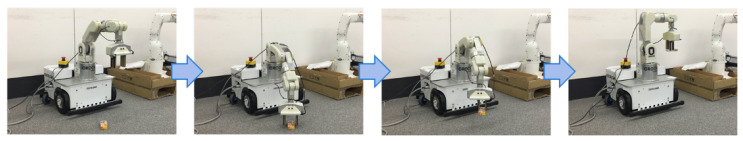
Example of grasping with different hardware.

**Figure 13 sensors-21-03804-f013:**
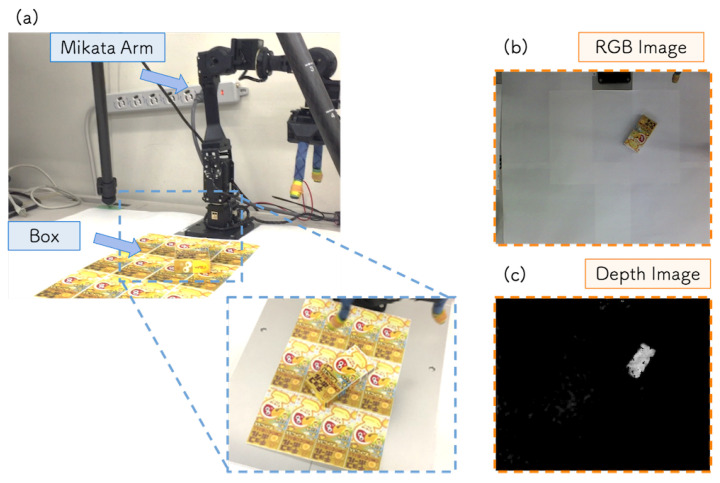
(**a**) Experimental environment at the TE step for the robot grasping application with an RBG-D camera. (**b**,**c**) RGB image and depth image collected at the DC step.

**Figure 14 sensors-21-03804-f014:**
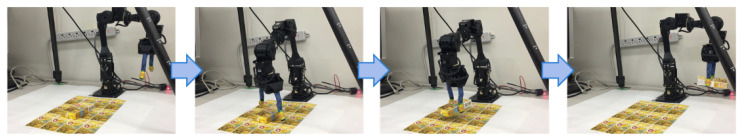
Example of grasping using RGB-D camera data.

**Figure 15 sensors-21-03804-f015:**
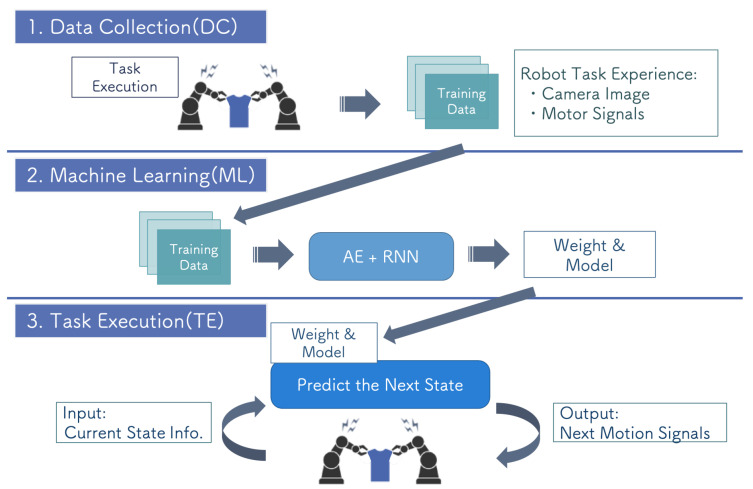
System flow overview of end-to-end robot motion learning.

## Data Availability

Not applicable. However, we published an operation manual for the robotic grasping system based on our flow on a web page. All software used in the DC, ML, and TE steps can be installed from the links at https://ogata-lab.jp/technologies/development-of-a-basic-educational-kit-for-robot-development-using-deep-neural-networks.html.

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
