# Peer review of "Development of a Basic Educational Kit for Robotic System with Deep Neural Networks"

_sensors, 2021, doi:10.3390/s21113804_

Round 1
Reviewer 1 Report
The paper proposes an educational kit for robotic system development with deep neural networks (DNNs) aimed to educate beginners in both robotics and machine learning, especially the use of DNNs. To evaluate the practicality of the proposed system flow, the authors implemented it for a physical robotic grasping system using robotics middleware.
Comments:
- The paper must explicitly state its novelty and contribution at the end of the Introduction section.
- There is no analysis of related works on educational robotics. Discuss related works on using educational robots for project based teaching, STEM, collaborative learning, etc. For example, you can discuss “Educational Robotics: Platforms, Competitions and Expected Learning Outcomes”, “Educational robots as collaborative learning objects for teaching computer science”, “Educational robots for internet-of-things supported collaborative learning”, among others. Summarize the limitations of the previous works as a motivation for your study.
- Three requirements formulated in Section 2.2 are too vague and generic. You should be able to verify how your robotic system satisfies the requirements, which does not seem possible now. Also since you propose an educational robotics system, you should include at least one pedagogical requirement/goal.
- In Lines 114-117, you also should evaluate and discuss the design alternatives.
- There is not much information about the methodology related to the research. There is a gap between the formal description of the method and its practical application for education.
- Present and discuss some example pedagogical scenarios of how the developed robotic system could be used in a classroom for teaching/learning.
- The educational/pedagogical usability of the developed robotic system has not been evaluated. See, for example, “Applying Pedagogical Usability for Designing a Mobile Learning Application that Support Reading Comprehension”, how to do that.
- Can you present some numerical results of experiments performed with grasping to demonstrate that the developed system works?
- Present a critical discussion section and discuss the limitations of robots for education as well as any threats to the validity of the results. Make a connection between your results and what were the results of previous similar studies, and the new and unique thing that you created. Also explicitly say what we can learn from your work.
- Conclusions should be improved as it lacks insights and recommendations for further research in this domain.
Reviewer 2 Report
The idea of the authors is valuable. However, this writing did not successfully materialize it. Some comments that support this phrase are the following:
1. It seems to me that there is a profound confusion between the term Educational and Implementation. This text presents an implementation since it does not transmit new knowledge. It is only a procedure to start a task.
2. The authors' tasks to educate on the subject are highly complex since, in addition to using the concept of Deep Neural Networks, they use Robotics and Computer Vision. The recommendation is to use straightforward problems like The MNIST database of handwritten digits and develop a didactic way of explaining how a deep Neural network architecture works.
3. The text presented is insufficient to consider that it met the three requirements of point 2.2. Authors must completely redesign their material.
4. It is clear that the article does not present a scientific contribution, but neither is it a didactic one. If the authors modify the material to fulfilled the second option, it could be considered for publication.
Reviewer 3 Report
I have read manuscript with great attention and interest. The article deals with a topic of education system for deep neural networks and robotics. The application is interesting and I think that the article can be accepted after minor revision. In general, you have a proper academic way of referring and a good language. I really like the introduction and state of the art sections.
Congratulations to the authors of the work.
Comments and suggestions:
1. Why did you choose RT-middleware? Why do you not prefer Robot Operating System? You should argue your choice in the article.
2. What are the main limitations of your education system?
After revision, the article can be accepted.
Round 2
Reviewer 1 Report
The paper was carefully revised and the quality of the paper has improved. I recommend the paper to be accepted.
Reviewer 2 Report
Although the authors indeed made several corrections to the original text, the new information is focused on convincing the reviewer of their contributions and not on presenting the material to be helpful as a Basic Educational Kit.
Authors are invited to thoroughly rephrase the information they present so that the writing truly serves to provide readers with basic education on the subject.
